# Stopping the Mobile Robotic Vehicle at a Defined Distance from the Obstacle by Means of an Infrared Distance Sensor

**DOI:** 10.3390/s21175959

**Published:** 2021-09-05

**Authors:** Frantisek Klimenda, Roman Cizek, Matej Pisarik, Jan Sterba

**Affiliations:** Faculty of Mechanical Engineering, University of Jan Evangelista Purkyne in Usti nad Labem, Pasteurova 1, 400 96 Usti nad Labem, Czech Republic; cizek@outlook.cz (R.C.); matej.pisarik@post.cz (M.P.); jan.sterba@ujep.cz (J.S.)

**Keywords:** Robotino^®^ 4. generation, infrared distance sensor, Robotino^®^ View interface, AGV

## Abstract

The article deals with the creation of a program for stopping an autonomous robotic vehicle Robotino^®^ 4. generation at a defined distance from an obstacle. One of the nine infrared distance sensors located on the frame of the robotic vehicle in the front part of the frame is used for this application task. The infrared distance sensor characteristic is created from the measured experimental data, which is then linearized in the given section. The main aim of the experiment is to find such an equation of a line that corresponds to the stopping of a robotic vehicle with a given accuracy from an obstacle. The determined equation of the line is applied to the resulting program for autonomous control of the robotic vehicle. This issue is one of the many tasks performed by AGV in the industry. The introduction of AGVs into the industry is one of the many possibilities within Industry 4.0.

## 1. Introduction

At present, there is an increasing emphasis on a continuous production flow in individual industries. These industries include, for example, the automotive, chemical, pharmaceutical, food and many others. Within Industry 4.0, instead of manual or inflexible mechanical solutions (e.g., Forklift trucks, conveyors and others), the ones used are the so-called AGVs (Automated Guided Vehicle) trucks. These trucks help increase productivity and reduce costs associated with the internal logistics system, ensuring an efficient material flow. When implementing the AGV, it is important to pay attention to the coordination of the vehicle fleet. The vehicle routing system is responsible for calculating trajectories that minimize the total distance traveled by the AGV, considering various constraints such as the carrying capacity of each vehicle and the layout of the race where vehicles can orbit, while ensuring collision-free routes [1,2,3,4,5,6].

Previously robot systems were restricted to a stationary position. Mobile robots represent the next step in the development of robotics in that they can execute the same tasks as their stationary predecessors but, in addition, can move away from a position. This provides the prerequisites for dealing with countless additional tasks.

The Mars Pathfinder was launched 4 December 1996 and landed on Mars’ Ares Vallis on 4 July 1997. It was designed as a technology demonstration of a new way to deliver an instrumented lander and the first-ever robotic rover to the surface of the red planet. Pathfinder not only accomplished this goal but also returned an unprecedented amount of data and outlived its primary design life [7].

The motivation behind the development and analysis of mobile robots is largely due to the necessity and desire to use robots that operate with and for people in their daily environment in offices, hospitals, museums, libraries, supermarkets, sports facilities (lawn mowing), exhibition halls, airports, railway stations, universities, schools and, eventually, also in domestic use [8,9].

Automated guided vehicle systems can be found increasingly in use in production plants and hazardous areas. These are mobile robots that are floor-bound; in other words, a driverless conveyance system moving along the floor. The automatic tracking either runs along predefined lanes or freely definable routes within a store or factory premises.

In automatic lane tracking, the robotic vehicle moves along a predetermined path. The path can be created using a black line, where the robotic vehicle follows the line either with an infrared sensor or with a camera. The second way of moving along the line is that the line is created with a metal strip. The robotic vehicle then monitors the metal strip using a capacitive sensor. When the vehicle moves freely without following the lane, the robotic vehicle scans the place where it is to move and chooses the path from point A to point B itself. In both cases, the vehicle is equipped with distance sensors. As you approach an obstacle, if you follow the lane, the vehicle will stop until the obstacle is removed from the track. In the case of free movement of the vehicle without a lane, the vehicle is looking for an alternative way to go around the obstacle. If the obstacle cannot be bypassed, the vehicle stops until the obstacle is removed [10].

When using autonomous robotic vehicles in industry, it is problematic to set the input parameters for stopping the vehicle at a predetermined distance from the object (obstacle). Several parameters affect the stopping of the vehicle at a defined distance from the object. These parameters include the speed of the vehicle, the weight of the vehicle and the coefficient of adhesion between the wheels and the floor. The solution to this problem arose on the basis of the demand of industrial enterprises, which turned to us with the problems of setting appropriate parameters when creating programs for stopping autonomous vehicles at a given distance from the object. The individual parameters were determined experimentally, which can be suitably inserted into the program for the creation of an autonomous control of the robotic vehicle Robotino^®^ 4. generation.

## 2. Mobile Robot Robotino^®^ 4. Generation

The introduction of the article describes robotic vehicles and their applications from history to the present. We deal with the robot vehicle Robotino^®^ 4. generation (Figure 1), which is a mobile robot system produced by Festo Didactic. Robotino^®^ is the mobile robot platform for research and education. With its omnidirectional drive, sensors, interfaces and application-specific extensions, Robotino^®^ can be used very flexibly. The most important programming languages and systems are available for programming individual applications. For example, graphical programing interfaces are the Robotino^®^ View, Factory and LabVIEW. Programing languages are MATLAB/Simulink, C++ and JAVA [11]. In our case, we used the graphical programing interface Robotino View. The basic parameters of the Robotino^®^ 4. generation are shown in Table 1.

The Robotino^®^ View software not only enables trainees to program the behavior of the system, but also to modify and test it interactively online via WLAN. The mobile robot consists of several subsystems:Control subsystem;Drive subsystem;Sensors;Supply subsystem;Software subsystem.

Robotino^®^ has a total of nine infrared sensors arranged at an angle of 40° to each other around the base (Figure 2). Each distance sensor supplies a voltage value in volts whose magnitude depends on the distance to a reflecting object. The infrared distance sensor GP2Y0A41SK0F allows accurate relative distance measurements of an object between 40 mm and 300 mm. The electro-optical characteristics are shown in Table 2.

## 3. Results

The solution to the problem is focused on setting such conditions that the robotic vehicle in autonomous control stops from an obstacle at a specific distance. The boundary conditions for the experimental solution are the same conditions as in the real conditions in the industry. In practice, these vehicles are used in industrial buildings such as robotic supply vehicles and it is important to know how to program these vehicles to meet the needs of users. The aim of the solution is to choose the optimal parameters for the given application. First of all, it is important to know on which surface the vehicle will move, due to the coefficient of adhesion between the wheels and the floor cover. This parameter is one of the parameters determining the braking distance. The braking distance is related to the speed of the moving robotic vehicle. The speed in practice depends on the so-called Cycle Time and the distance of the transport of the load from the starting point to the end point. Cycle Time means how long it takes for the load to be transported from place A to place B in order not to stop, e.g., assembly on the line. Another parameter that affects the stopping distance from the obstacle is the type of sensor that is used. Each sensor works on a different principle and each sensor has a different signal response. The weight of the truck also has an effect on the braking distance., else inertia forces are for a truck without a load and else with a load. All these parameters should be included in the creation of the program for the movement of the autonomous robotic vehicle. The results of this work are optimal parameters that are used in practice for the given application.

The boundary condition of the assignment is the stopping of the robotic vehicle at a distance of 80 mm from the obstacle. Where did the distance of 80 mm from the obstacle come from? In this case, the Robotino^®^ 4. generation is equipped with a ramp with a plate. The plate has a diameter 80 mm larger than the diameter of the vehicle itself. To prevent the vehicle not crashed into the ramp when approaching a loading/unloading ramp, it must stop at a distance of 10–40 mm from the ramp. For these reasons, the stopping distance of the robotic vehicle from the ramp is set to 80 mm.

In this case, only one infrared distance sensor (the first sensor with position 0° on Figure 2) is used to stop the robotic vehicle accurately from the obstacle). The reason for using an infrared distance sensor is that the robotic vehicle is already equipped with nine infrared distance sensors around its perimeter. First of all, the characteristic of the infrared distance sensor was determined by experimental measurements. The result of the experiment is the dependence of the output voltage on the distance from the object for the infrared distance sensor. The initial value was 40 mm and the final value being 300 mm. The distance was always increased by 10 mm from the initial value. The velocity of the robotic car in all cases was 50 mm/s. Each measuring point was measured 10 times. At the end of the measurement, the average value at each point was calculated from the measured values (Table 3). Figure 3 shows a program of this experiment for the first measuring conducted in the Robotino^®^ View interface.

The program was created as a block diagram. First of all, three “Robotino’s motors” marked one, two and three (see on the right) were inserted into the program. The Robotino^®^ 4. generation has three motors rotated 120° relative to each other. The motors were connected to the “Omnidrive (inverse)” function block. This function block calculates *vx*, *vy* and *omega* from the motors’ rotation speeds. Parameters from Omnidrive entered to the “Multiplication” function block. The number of outputs was set in this function block. The maximum output was 10. Two outputs were set for this application. The first was the speed, e.g., 50 mm/s. The second output was a subassembly of the program. This subassembly consisted of the “Greater” function block (the Output is true, if Input 1 is greater Input 2). The function blocks “Scale” (scaling of values) and “Distance sensor 1” (infrared distance sensor at position 0° on Figure 2) were connected to Input 1. In the “Scale” function block, we entered the values of the equation of the line (3). The value of the distance from the obstacle, e.g., 4 cm, was entered into Input 2. The unit “cm” was used because the infrared distance sensor senses the distance in “cm”. The result of the program was that the robotic vehicle moved at a speed of 50 mm/s in a given direction. The infrared distance sensor continuously senses the distance in front of it. If the distance is equal to or less than 4 cm, the vehicle will stop.

Figure 4 shows the characteristic of the infrared distance sensor from measured values.

From the characteristic of the infrared distance sensor, the following was visible in the Power Trendline format:*y* = 82.075 · *x*^−0.927^.(1)

The Power Trendline needed to be converted to a Linear Trendline. We used the equation of the line for the conversion:*y* = *k* · *x + q*,(2)
where: *k*—straight line directive; *q*—displacement along the *y*-axis.

As a condition for stopping the robotic vehicle at a distance of 80 mm from the obstacle, it was necessary to linearize the section near this value. The section from 50 mm to 100 mm was selected. From Table 3 into Equation (2), a value was assigned corresponding to the voltage at distances 50 and 100 mm. The result was the equation of the line in the form of:*y* = −5.32 · *x +* 16.33.(3)

The parameters from the equation of line (3) were inserted into the function in the program—see Figure 5—and, thus, the infrared distance sensor was linearized.

The final program was applied to Robotino^®^ 4. generation. The measurement was performed a total of 10 times. As a result, the mean value, including the standard deviation, was calculated
*y* = 80 ± 5 mm.(4)

## 4. Discussion

When manually controlling the robotic vehicle, the communication between the operator and the robotic vehicle is delayed due to the delay of Wi-Fi communication. This delay depends on the data rate of the Wi-Fi network and is in the order of milliseconds. However, manual control is not used in practice for autonomous movement of the robotic vehicle. Manual control is only used for a maintenance vehicle or to create a program for autonomous control. Another effect on accuracy is the communication between the sensor and the control unit. This accuracy depends on the type and kind of sensor. For example, the infrared distance sensor GP2Y0A41SK0F used by us had a response of 16.5 ms ± 3.7 ms. The braking distance of the robotic vehicle depends on the speed of movement of the robotic vehicle. After previous experimental measurements, it was found that the relationship between the braking distance and the speed of the robotic vehicle is almost linear. For example, at a speed of 200 mm/s, the distance of the braking distance may be almost half as long as the required stop from the obstacle. The experimental measurements were measured at a maximum load capacity of the robotic vehicle of 30 kg and on a floor with a smooth concrete surface, where the coefficient of friction between the rubber wheel and the smooth dry concrete was in the range of 0.6–0.85. The coefficient of friction between the wheel and the surface on which the robotic vehicle moves is another limiting parameter on the distance of the vehicle stop from the obstacle. The smoother the surface, the longer the braking distance of the vehicle and, conversely, the rougher the surface, the shorter the braking distance. Smooth dry concrete, on which experimental measurements were performed, is the most common floor surface in industrial halls.

The experimental solutions were performed with the new Robotino^®^ 4. generation robotic vehicle. The problem that will be solved in further research is the wear of individual parts of the vehicle. In older vehicles, rotating parts and bearings of rotating parts are most often worn due to almost continuous operation. Furthermore, the rubber wheels of the vehicle wear out, which has an effect on the braking distance and the safe operation of the vehicle. In practice, it was found that an exceeded load capacity of the vehicle, on the one hand, wears out the rotating parts—as already mentioned—, but mainly it prolongs the braking distance; thus, causing the vehicle to stop inaccurately from an obstacle. This issue is currently being addressed with industrial companies that use the Robotino^®^ 4. generation robotic mobile vehicle. Furthermore, the effect of local changes in adhesion on the floor, such as puddles of water, oil, sand and others, will be experimentally addressed.

The issue of the precise stopping of a robotic vehicle in front of an obstacle at a given distance in an industrial sector has not been addressed in the available literature. In the available literature, avoiding an obstacle is the most addressed. Ultrasonic sensors are most often used to detect an obstacle. 

In their work, Haltorp and Bredhe [13] focus on the construction of a three-wheeled robotic vehicle, which is equipped with four ultrasonic sensors of type HC-SR04. One sensor is in the front, the other sensor is in the rear and the other two sensors are on the left and right side of the vehicle. This division covers almost the entire circumference of the vehicle frame. The sensors can identify a distance from 20 to 4000 mm with an accuracy of 3 mm. Ultrasonic sensors emit short, high-frequency sound pulses at regular intervals. These propagate in the air at the velocity of sound. If they strike an object, then they are reflected back as echo signals to the sensor, which itself computes the distance to the target based on the time-span between emitting the signal and receiving the echo. Using an experimental solution, they determined the stopping of the vehicle at the exact distance from the obstacle. The distance from the obstacle was 10, 50, 150 and 300 mm. The results were within tolerance except for a distance of 10 mm. At a distance of 10 mm, the result was out of tolerance, which is due to the fact that the minimum readable distance of the ultrasonic sensor is 20 mm. Unfortunately, in addition to the test distance, no other boundary conditions were mentioned in the work, which include the speed of the vehicle, the coefficient of adhesion between the tires and the floor cover. The tread pattern also affects the experiment, as each tire has a different braking distance.

Two ultrasonic sensors for obstacle detection were used in the design of an autonomous robotic vacuum cleaner by Cerny [14]. The purpose of these sensors was to detect an obstacle at a distance of 100–120 mm. At the obstacle in the first row, the vehicle stopped and then went around the obstacle to the left or right. The sensors were placed side by side at the front of the vehicle at such a distance that their detection beams did not cross. In this work, the author did not solve the exact stop from the obstacle, he solved only to avoid a collision with the obstacle.

Another example of the use of an ultrasonic sensor is in a publication by Cechmanek [15]. In his work he describes the use of four ultrasonic sensors for the autonomous movement of a robotic vehicle. The sensors are located on each side of the truck. Compared to the previous examples, the vehicle is extended by an LCD display, where the measured distances from the obstacle in individual directions are recorded. In his contribution, not solving a stop at a specific distance from an obstacle, only if the vehicle approaches an obstacle at a distance shorter than 300 mm, will change its direction so that the vehicle avoids the obstacle.

In conclusion to the discussion, it can be stated that ultrasonic sensors for measuring the distance from an obstacle are more used in practice.

## 5. Conclusions

The paper describes one of the possibilities of using the sensor to stop the mobile robot truck Robotino^®^ 4. generation from the company Festo Didactic. This robotic vehicle is not only used in the education of students in the field of automation and robotics, but especially in industry. In industry, it is used mainly as an autonomous supply robotic vehicle.

The main advantage of this vehicle is its multifunctionality. The vehicle can not only scan the space in which it moves and choose the optimal route of movement from the starting point A to the end point B, but above all its safety during operation thanks to the sensors with which it is equipped is paramount. The vehicle has a Bumper around its perimeter, which immediately stops the vehicle when in contact with the object, as well as nine sensors distributed around the perimeter of the frame by 40°, for which the distance value can be set from 40 to 300 mm. This guarantees not only safety in terms of a collision with an obstacle, but also a possible collision with a person if they move into the path of a mobile vehicle. The safety of people moving around autonomous robotic vehicles is paramount. People should consider moving autonomous robotic carts and keep a safe distance from the cart path. There may be situations where a person is entangled in the path of an autonomous robotic vehicle. In this case, the distance sensors must be set so that the truck stops as soon as possible or selects a different path to avoid contact with humans. The distance between the sensors must be set with regard to the speed of the truck and the type of surface on which the truck is moving. The higher the speed of the vehicle and the smoother the surface, the greater the distance of the sensor must be set so that, as already mentioned, the vehicle does not collide with humans.

The result of the input solution, where it is required to stop the vehicle 80 mm from the obstacle, is to create a program for autonomous control of the vehicle and its stopping from the obstacle by means of a distance sensor. Due to the negative effects that limit the exact stopping from an obstacle—see the last paragraph in the previous chapter—the truck stops at a distance of 80 ± 5 mm. This tolerance is acceptable in the industry.

An example of the use of a robotic vehicle is one of the possibilities of using automation and robotics within Industry 4.0, which is becoming more and more widespread.

The next step will be an experimental solution to the effect of a local change in adhesion between the wheel and the surface, e.g., puddles of water, oil, sand and others. The robotic vehicle does not evaluate these parameters as an obstacle, but in reality, it has an impact on the operational safety of the robotic vehicle.

## Figures and Tables

**Figure 1 sensors-21-05959-f001:**
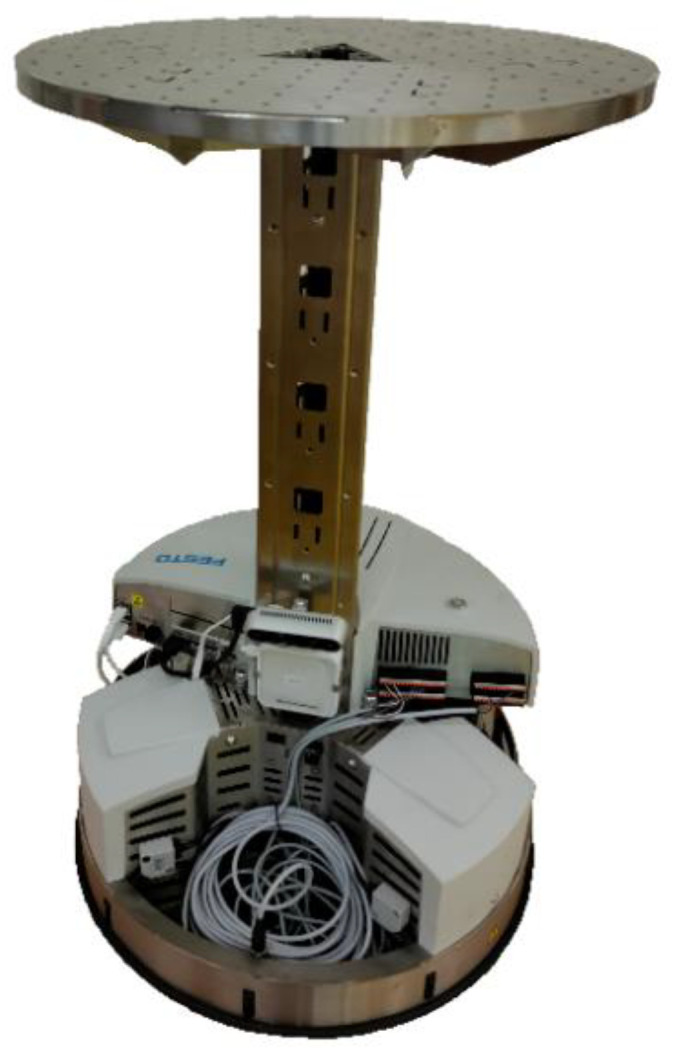
Robotino^®^ 4. generation.

**Figure 2 sensors-21-05959-f002:**
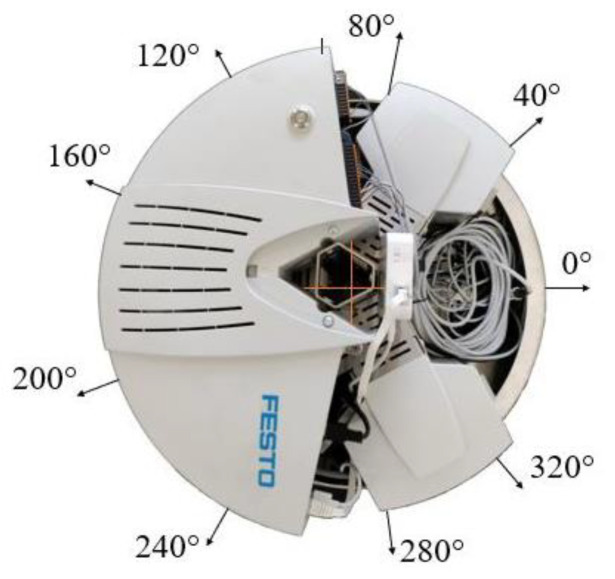
Infrared distance sensors arranged around the base.

**Figure 3 sensors-21-05959-f003:**
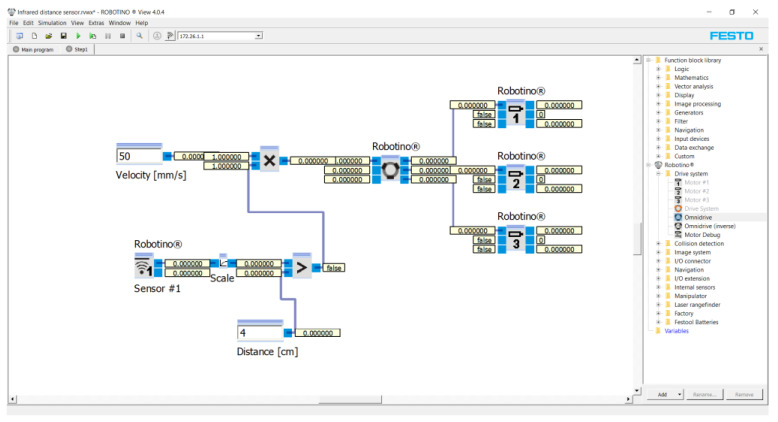
Program for the first measuring conducted in the Robotino^®^ View interface.

**Figure 4 sensors-21-05959-f004:**
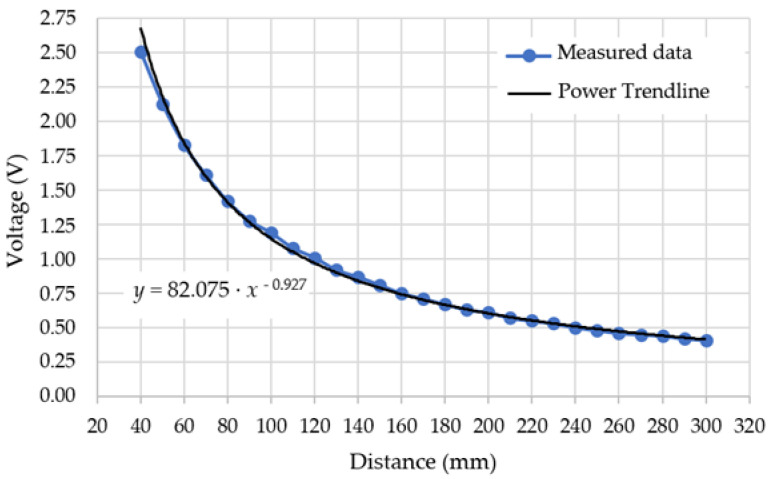
The characteristic of the infrared distance sensor from the measured values.

**Figure 5 sensors-21-05959-f005:**
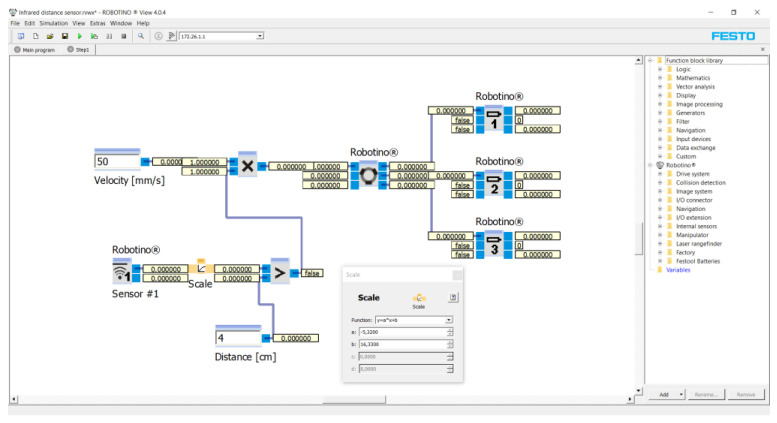
The final program for stopping at a distance from the obstacle conducted in the Robotino^®^ View interface.

**Table 1 sensors-21-05959-t001:** Basic parameter of the Robotino^®^ 4. generation [12].

Parameter	Value
Height	325 mm
Diameter	450 mm
Total weight (curb weight)	20 kg
Total weight (including 4 battery packs)	22.8 kg (ca. 700 g each accumulator)
Maximum payload	30 kg (centered)
IP protection	IP 00
Battery voltage	18 V
Housing material	Stainless steel, PA6
Degrees of freedom	3 (translator in the *x* and *y* direction, rotational about the *z* axis)

**Table 2 sensors-21-05959-t002:** The electro-optical characteristics of infrared distance sensor GP2Y0A41SK0F [12].

Parameter	Symbol	Conditions	Min.	Max.
Measuring distance range	Δ*L*	(Note 1)	40 mm	300 mm
Output terminal voltage	*V_o_*	*L* = 300 mm (Note 1)	0.25 V	0.55 V
Output voltage difference	Δ*V_o_*	Output change at *L* change (300 mm–>40 mm) (Note 1)	1.95 V	2.55 V
Average supply current	*I_cc_*	*L* = 300 mm (Note 1)	-	22 mA

(Note 1)—Using reflective object: white paper.

**Table 3 sensors-21-05959-t003:** Measured values.

Distance (mm)	Voltage (V)	Distance (mm)	Voltage (V)
40	2.51	180	0.67
50	2.13	190	0.63
60	1.83	200	0.61
70	1.61	210	0.57
80	1.42	220	0.55
90	1.28	230	0.53
100	1.19	240	0.50
110	1.08	250	0.48
120	1.01	260	0.46
130	0.92	270	0.45
140	0.87	280	0.44
150	0.81	290	0.42
160	0.75	300	0.41
170	0.71	-	-

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
