# Peer review of "Stopping the Mobile Robotic Vehicle at a Defined Distance from the Obstacle by Means of an Infrared Distance Sensor"

_sensors, 2021, doi:10.3390/s21175959_

Round 1

Reviewer 1 Report

The paper's composition is coherent; the structure is logical and meets the goal of the paper. The title "Stopping the mobile robotic vehicle at a defined distance from the obstacle by means of an infrared distance sensor" puts well the paper's objective; it is clear and expresses the issue being assessed very well. The results have a certain scientific value and therefore it may be interested both for journal readers as well as other professional communities. The abstract is formulated adequately along with the true picture of the paper. Conclusions are related to the results presented before reflecting the assessed issue at a professional level. I found the paper well-written and cohesive. Authors appear to be professionals, very well oriented and involved in the observed issue. The length of the paper is adequate to the significance of the topic. However, major revision would be necessary to get the manuscript published in the journal. It is recommended that the authors make a relatively major revision, and the specific amendments to the text are as follows: 

  • The Discussion section is recommended to be set aside from the Results section. Some kind of polemic discourse comparing the research outcomes with the literature overview part would be beneficial to be involved in Discussion. Please explain in Literature overview and Discussion sections how and in what way the stopping the mobile robotic vehicle issue is widening the subject area compared with other published material. 
  • The goal explicitly stated within the Introduction clearly expressing the main problem and purpose of the paper and author's intention being assessed and discussed within the paper along with its clear and unambiguous formulation are required to be proposed. Please specify in the Introduction section the main question addressed by the research, its potential to be used in action and added value. Please specify the originality and relevance of the topic in the field of Industry 4.0. 
  • In introduction to underline the added value, novelty, purpose along with the brief methodology and data sources being used and ways of application of the research results would be recommended. Regarding the methodology please try to be focused on detailed step by step approach, also a hypothesis statement would be appreciated. 
  • I recommend adding to the Conclusion section authors’ further research directions within this explored issue along with a brief research limitation. The final research statements in Conclusion should be more supported by your evidence and arguments from your own research findings. 
  • Please add more sources within the references especially from domestic academic and professional area.  
  • Finally, please include any additional comments on the tables and figures to describe them in more details and link them to each other to express their connection and progression within the research process.

Author Response

Dear Reviewer,

I incorporated your comments into the article.
I am sending you brief answers to your questions in the attachment.

Best regards

Frantisek Klimenda

Reviewer 2 Report

Authors should further clarify and elaborate novelty in their contribution.

What are the limitations of the present work?

What are some key issues that present study has addressed?

What are the practical implications of this research?

Elaborate this line “Safety for autonomous robotic vehicles comes first” line, 143. 

Lines, 123- 128, authors mentioned the different delays. Explain in detail what are the effects of such delays on the performance and efficiency of robotic vehicle.

Why present work used the infrared sensors?

Line 83, How authors get the 80mm distance?

Explain how the differentiation is done between line-bound and line-free tracking? 

Author Response

Dear Reviewer,

I incorporated your comments into the article.
I am sending you  answers to your questions in the attachment.

Best regards

Frantisek Klimenda

Round 2

Reviewer 1 Report

The revised paper titled “Stopping the mobile robotic vehicle at a defined distance from the obstacle by means of an infrared distance sensor“ intended to be published in Sensors Journal meets all the requirements for a professional scientific journal. All the significant comments, recommendations and remarks of reviewers have been incorporated into the manuscript in a proper way giving the paper higher added value and professional features. 

Reviewer 2 Report

.